# Interventions for Shiga toxin-producing *Escherichia coli* gastroenteritis and risk of hemolytic uremic syndrome: A population-based matched case control study

Shota Myojin[1,2], Kyongsun Pak[3], Mayumi Sako[4], Tohru Kobayashi[5], Takuri Takahashi[6], Tomimasa Sunagawa[6], Norihiko Tsuboi[7], Kenji Ishikura[8], Masaya Kubota[9], Mitsuru Kubota[10], Takashi Igarashi[11], Ichiro Morioka[2], Isao Miyairi[1,12]*

1 Division of Infectious Diseases, Department of Medical Subspecialties, National Center for Child Health and Development, Tokyo, Japan, 2 Department of Pediatrics and Child Health, Nihon University School of Medicine, Tokyo, Japan, 3 Biostatistics Unit, Department of Data Science, Clinical Research Center, National Center for Child Health and Development, Tokyo, Japan, 4 Department of Clinical Research Promotion, Clinical Research Center, National Center for Child Health and Development, Tokyo, Japan, 5 Department of Data Science, Clinical Research Center, National Center for Child Health and Development, Tokyo, Japan, 6 Infectious Disease Surveillance Center, National Institute of Infectious Diseases, Tokyo, Japan, 7 Division of Critical Care Medicine, Department of Critical Care and Anesthesia, National Center for Child Health and Development, Tokyo, Japan, 8 Department of Pediatrics, Kitasato University School of Medicine, Kanagawa, Japan, 9 Division of Neurology, Department of Medical Subspecialties, National Center for Child Health and Development, Tokyo, Japan, 10 Department of General Pediatrics & Interdisciplinary Medicine, National Center for Child Health and Development, Tokyo, Japan, 11 National Center for Child Health and Development, Tokyo, Japan, 12 Department of Pediatrics, Hamamatsu University School of Medicine, Shizuoka, Japan

* miyairi@hama-med.ac.jp

**Data Availability Statement:** All relevant data are within the paper and its Supporting Information files.

## Abstract

### Background

The role of antibiotics in the treatment of Shiga toxin-producing *Escherichia coli* (STEC) infection is controversial.

### Objectives

To evaluate the association between treatment (antibiotics, antidiarrheal agents, and probiotics) for STEC infection and hemolytic uremic syndrome (HUS) development.

### Patients and methods

We performed a population-based matched case-control study using the data from the National Epidemiological Surveillance of Infectious Diseases (NESID) between January 1, 2017 and December 31, 2018. We identified all patients with STEC infection and HUS as cases and matched patients with STEC infection without HUS as controls, with a case-control a ratio of 1:5. Further medical information was obtained by a standardized questionnaire. Multivariable conditional logistic regression model was used.

**Funding:** This work was supported in part by a grant from the Ministry of Health, Labour and Welfare, Japan (H30-Shinko-003). The funders had no role in study design, data collection and analysis, decision to publish, or preparation of the manuscript.

**Competing interests:** The authors have declared that no competing interests exist.

## Results

7760 patients with STEC infection were registered in the NESID. 182 patients with HUS and 910 matched controls without HUS were selected. 90 patients with HUS (68 children and 22 adults) and 371 patients without HUS (266 children and 105 adults) were included in the main analysis. The matched ORs of any antibiotics and fosfomycin for HUS in children were 0.56 (95% CI 0.32–0.98), 0.58 (0.34–1.01). The matched ORs for HUS were 2.07 (1.07–4.03), 0.86 (0.46−1.61) in all ages treated with antidiarrheal agent and probiotics.

## Conclusions

Antibiotics, especially fosfomycin, may prevent the development of HUS in children, while use of antidiarrheal agents should be avoided.

## Introduction

Shiga toxin-producing *Escherichia coli* (STEC) may cause hemorrhagic colitis and hemolytic uremic syndrome (HUS), which is characterized by microangiopathic hemolytic anemia, thrombocytopenia, and renal insufficiency [1]. Annually, STEC is estimated to cause 2.8 million acute illnesses worldwide, leading to 3890 cases of HUS [2]. As the leading cause of pediatric acute kidney failure in most developed countries [3], the mortality rate of STEC infections range from 3% to 5% during the acute phase even in the modern era of medicine [4]. The estimated annual cost of STEC infections is more than US$ 400 million in the United States [5].

Several studies aiming to develop effective interventions have attempted to identify risk factors for the progression of STEC infection to HUS. Young age [6–10], female sex [9], the STEC O157 serotype encoding only Shiga toxin 2 [11, 12], disease severity such as vomiting and bloody diarrhea [9], and antimotility agents [13, 14] are reported as risk factors for progression to HUS. The effect of probiotics for STEC infections remains unknown [15] although it is frequently prescribed for gastroenteritis in Japan [16]. To date, avoiding antimotility drugs remains the only modifiable medical intervention.

Whether antibiotics should be administered to patients with STEC infection remains controversial although a number of reports and reviews in the bibliography have indicated that certain antibiotics including fosfomycin when utilized in appropriate timing can be beneficial to the outcome of this infection. The majority of studies, including a recent meta-analysis, have concluded that there is an association between antibiotics and increased risk of HUS [17] (odds ratio [OR] 2.24, 95% confidential interval [CI] 1.45–3.46). In contrast, a number of Japanese studies not included in the recent meta-analysis have demonstrated the protective effect of fosfomycin in patients with STEC infection [18–22]. These studies were unique as fosfomycin is not available in most other countries. However, there were shortcomings in the study design, precluding the inclusion of these studies as evidence to support the clinical use of fosfomycin [17]. Nevertheless, the use of antibiotics in patients with STEC infection is a common approach in Japan [19, 23].

Evidence supporting treatment intervention, especially the use of antibiotics for STEC infection will provide a proactive strategy in HUS prevention and may reduce morbidity associated with STEC infection. A properly designed population-based study capturing the entire population at risk for STEC-related HUS should resolve the shortcomings of previous studies in Japan. We therefore hypothesized that antibiotics, particularly fosfomycin, would reduce

the risk of HUS in patients with STEC infection. The aim of the present study was to evaluate the association between treatment for STEC infection and HUS development.

## Materials and methods

### Study design

This population-based matched case-control study examined the association between treatment for STEC infection and development of HUS in Japanese patients. Patients with STEC infection were identified from the database of National Epidemiological Surveillance of Infectious Diseases (NESID), a national infectious diseases surveillance system. STEC infections are a notifiable disease, and all cases are mandated to be reported to a regional public health center. During reporting, each public health center enters information related to the STEC infection via an online system in accordance with the Act on Prevention of Infectious Diseases and Medical Treatment for Patients with Infectious Diseases. The registered data are evaluated and summarized in the Infectious Diseases Surveillance Center of the National Institute of Infectious Diseases (NIID). The institutional review boards of the National Center for Child Health and Development (ethics reference number, 2019–043) and NIID (ethics reference number, 1065) approved this study. The investigators were granted access to limited non-personally identifiable information from the NESID. The requirement for informed consent was waived. An opt-out model was adapted, and the opportunity for patients to refuse inclusion in the study was maximized using a website and a poster which was discretionally displayed in each medical facility. All collaborating physicians could request the institutional review board of the National Center for Child Health and Development to deliberate and determine their participation.

### Data collection

Based on a priori power analysis, we considered that sufficient power could be secured to detect OR with clinical significance in all possible scenarios. Information extracted from the NESID database included data on patients with STEC infection and asymptomatic careers who were diagnosed between January 1, 2017 and December 31, 2018. Cases were selected from patients with a record of HUS diagnosis in the NESID. Controls were selected from patients without a record of HUS who were matched to cases by sex (male or female), age groups (0–6, 7–15, 16–64, and ≥65 years), and the presence of bloody stool (yes or no) with a case-control ratio of 1:5. All physicians who reported patients who were selected as cases and controls were contacted, and data were collected based on a standardized questionnaire on the clinical course. Patients fulfilling the following criteria were excluded: 1) there was no response by the treating physician, 2) physician's refusal to cooperate, 3) missing mandatory data (month and year of birth, presence of bloody stool, date of HUS diagnosis, prescribed antibiotics, and antibiotic prescription date), 4) use of inadequate diagnostic criteria for HUS (cases only), 5) diagnosis of HUS (controls only), 6) asymptomatic patients (controls only).

### Variables

Information on the following variables were collected: demographic data (month and year of birth, body weight, information on referrals and hospitalization, medical history, and drug history), presence of specific symptoms (vomiting, diarrhea, abdominal pain, fever, and bloody stool) and their onset, initial and worst values for laboratory parameters (white blood cell count [WBC], hemoglobin, platelet count, C-reactive protein [CRP], blood urea nitrogen, serum creatinine, serum sodium concentration, aspartate aminotransferase and alanine aminotransferase), diagnostic information on STEC infection (stool culture positivity, type of

toxin [Shiga toxin 1 and 2], serotype [O157, O26, O103, O111, and others], and serum anti-Shiga toxin antibody), diagnostic information on HUS (date of diagnosis, results and dates of tests used for HUS diagnosis, and presence of schistocytosis), treatments (antibiotics, antidiarrheal agents, probiotics, and dialysis), and outcomes (final outcome, complications including encephalopathy, and date of last visit).

### Outcome, exposures, and potential confounders

The primary outcome was development of HUS after STEC infection. The HUS diagnosis was based on the presence of microangiopathic hemolytic anemia (hemoglobin < 10 g/dL), thrombocytopenia (platelet count < 150 000 cells/μL), and renal insufficiency (creatinine level above the upper normal limit for age [1]). Exposures of interest were antibiotic administration (any antibiotic), antibiotic administration by type (fosfomycin, quinolones, macrolides, beta-lactams, and others), antidiarrheal agent administration, probiotic administration, and STEC serotype (O157 or not). In cases, treatments were counted only when they were administered before development of HUS.

Following variables were defined based on the information obtained using the questionnaire. Six region codes were created based on the physician addresses (Hokkaido/Tohoku, Kanto, Chubu, Kinki, Chugoku/Shikoku, and Kyushu/Okinawa). The leukocyte count and CRP level were converted into nominal variables according to cutoff values of 10 000/μL and 1.2 mg/dL, respectively, in accordance with a previous study [22]. Antibiotics, antidiarrheal agents, and probiotics were classified according to the World Health Organization Anatomical Therapeutic Chemical classification system [24].

### A priori power analysis

The National Institute of Infectious Diseases reported that there were 180 and 5006 patients with and without hemolytic uremic syndrome (HUS), respectively, among patients with Shiga toxin-producing *Escherichia coli* (STEC) infection between 2017 and 2018 [25, 26]. In a previous study evaluating the association between antibiotics and development of HUS in Japanese tertiary centers, the authors reported that approximately 60% (40 of 64) of patients with HUS and 80% (43 of 54) of patients without HUS among those with STEC infection received antimicrobial agents, with an antimicrobial exposure OR of 0.375 for the HUS group compared with the non-HUS group [18]. Power analysis based on these results indicated that 100 and 500 patients were necessary in the HUS (case) and non-HUS (control) groups, respectively. The assumed OR for the alternative hypothesis that antibiotics would reduce the risk of HUS was between 0.375 and 0.450. S1 Table in S1 File shows the power analysis of 100 cases and 500 controls, based on 75%, 80%, and 85% antimicrobial exposure rates in the non-HUS group, with a two-tailed test at a significance level of 5%. We considered that sufficient power could be secured to detect OR with clinical significance in all possible scenarios. The power analysis was performed with the statistical software R, Package epiR (version 1.0–2).

### Statistical analysis

In demographic comparisons between cases and controls, the chi-square and Student's *t* tests were used to compare differences in proportions and mean values, respectively. In the main analysis, univariable and multivariable conditional logistic regression models were applied to evaluate the association between treatments (any antibiotics, fosfomycin, quinolones, macrolides, beta-lactams, antidiarrheal agents, and probiotics) and development of HUS in all age groups, while taking stratification by matching factors into account. The reference exposure in each analysis was non-administration of each treatment of interest. Matched OR (mOR) and

adjusted mOR ($mOR_{adj}$) values with 95% CI values were reported. Covariates used for adjustment in multivariable models to evaluate the association of any antibiotics, fosfomycin, quinolones, macrolides, and beta-lactams were patient background characteristics (sex, age, and region), presence of bloody stool, laboratory results (WBC and CRP at initial presentation), serotype (O157), and medical interventions (antidiarrheal agents). For the analysis of antidiarrheal agents, the same covariates were used with one exception; the use of any antibiotics was included instead of the use of antidiarrheal agents as medical intervention. Covariates used in the analysis of probiotics were also the same with one exception; the use of antidiarrheal agents and any antibiotics was included as medical intervention instead of the use of probiotics. These covariates were chosen because sex [9], age [6–10], signs of disease severity such as bloody stool [9], and serotype O157 [11, 12, 27] have been reported to be associated with HUS development. Region was added as a covariate due to the possibility of local differences in clinical practice across Japan. Initial WBC count and CRP level were used as covariates because they could be used by physicians to assess disease severity, potentially influencing their decision for antibiotic prescription. Additionally, subgroup analyses were performed in children (0–15 years of age) and adults (≥16 years of age), and patients with confirmed infection by O157 serotype, given that young age and O157 serotypes are known risks for developing HUS.

All data were analyzed using IBM SPSS statistical software (version 26.0, IBM, Tokyo Japan).

## Results

From January 1, 2017 to December 31, 2018, we identified 7760 patients with STEC infections, including 182 (2.3%) patients diagnosed with HUS, in the NESID. The present study included these 182 patients with HUS as well as 910 patients without HUS. After retrieving questionnaire results and confirming eligibility, 92 cases fulfilling the following criteria were excluded from the study: no response (66 [36.3%]), physician's refusal to cooperate (16 [8.8%]), failure to fulfil the precise diagnostic criteria (5 [2.7%]), and missed mandatory data (5 [2.7%]). A total of 539 controls fulfilling the following criteria were excluded from the study: no response (384 [42.2%]), physician's refusal to cooperate (106 [11.6%]), diagnosis of HUS (12 [1.3%]), asymptomatic carrier (33 [3.6%]), and missed mandatory data (4 [0.4%]). Therefore, 90 (49%) cases and 371 (41%) controls were included in the final analyses (Fig 1).

### Baseline characteristics of the cases and controls

Table 1 shows the baseline characteristics of patients who were registered in the NESID, matched controls, and cases and controls included in the final dataset. Although about half of the cases were not included mostly due to lack of responses, the proportions of sex, age, and region of patients with HUS in the analyzed dataset were similar to those of patients with HUS registered in the NESID. The baseline characteristics of the controls in the analyzed dataset were generally consistent with those of the controls without HUS after matching. In the analyzed dataset, the cases were significantly more likely to report vomiting, fever, and severe bloody stool (all $P < .001$). Similarly, the frequency of Shiga toxin 1 detection was significantly lower in cases than in controls ($P < .001$). Serotype O157 was significantly more frequently detected in cases than in controls ($P < .001$), and no patient with serotype O26 developed HUS ($P < .001$). The rate of complications and death were significantly higher in cases than in controls ($P < .001$ and.007, respectively) (Table 2). Among the laboratory data, the values for initial and worst WBC counts, CRP, blood urea nitrogen, creatinine, aspartate aminotransferase, and alanine aminotransferase were higher in cases compared to controls (Table 3). In contrast, initial and worst hemoglobin levels and platelet count were lower in cases compared to controls.

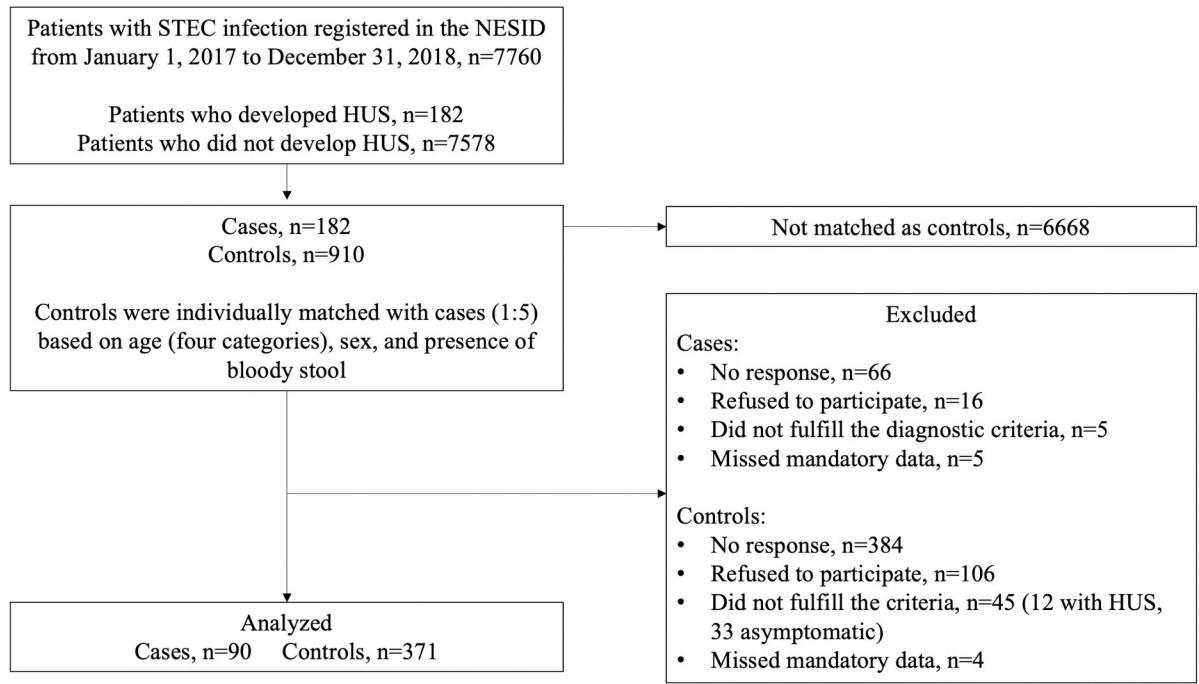

**Fig 1. Study flow chart.** HUS, hemolytic uremic syndrome; NESID, National Epidemiological Surveillance of Infectious Diseases; STEC, Shiga toxin-producing *Escherichia coli*.

**Table 1. Baseline characteristics of the NESID cohort, matched controls, and cases and controls included in the main analysis.**

| | Group, no. (%) | | | | |
|---|---|---|---|---|---|
| | NESID, n (%) | | Matched controls, n (%) | Analysis dataset, n (%) | |
| | HUS | Non-HUS | | Cases | Controls |
| All | 182 (100) | 7578 (100) | 910 (100) | 90 (100) | 371 (100) |
| Sex | | | | | |
| Female | 114 (62.6) | 4231 (55.8) | 570 (62.6) | 57 (63.3) | 213 (57.4) |
| Age, years | | | | | |
| 0–6 | 92 (50.5) | 1540 (20.3) | 460 (50.5) | 50 (55.6) | 196 (52.8) |
| 7–15 | 28 (15.4) | 871 (11.5) | 140 (15.4) | 18 (20.0) | 70 (18.9) |
| 16–64 | 41 (22.5) | 4173 (55.1) | 205 (22.5) | 18 (20.0) | 72 (19.4) |
| ≥65 | 21 (11.5) | 994 (13.1) | 105 (11.5) | 4 (4.4) | 33 (8.9) |
| Area | | | | | |
| Hokkaido/Tohoku | 18 (9.9) | 1179 (15.6) | 35 (3.8) | 8 (8.9) | 10 (2.7) |
| Kanto | 86 (47.3) | 2807 (37.0) | 98 (10.8) | 41 (45.6) | 43 (11.6) |
| Chubu | 31 (17.0) | 1340 (17.7) | 92 (10.1) | 19 (21.1) | 35 (9.4) |
| Kinki | 27 (14.8) | 893 (11.8) | 217 (23.8) | 11 (12.2) | 96 (25.9) |
| Chugoku/Shikoku | 4 (2.2) | 501 (6.6) | 113 (12.4) | 3 (3.3) | 64 (17.3) |
| Kyushu/Okinawa | 1 (8.8) | 858 (11.3) | 355 (39.0) | 8 (8.9) | 123 (33.2) |

HUS, hemolytic uremic syndrome; NESID, National Epidemiological Surveillance of Infectious Diseases. All patients with a record of hemolytic uremic syndrome diagnosis in the NESID were included as cases. For each case, controls were randomly selected at a ratio of 1:5 based on the information on age, sex, and presence of bloody stool. A standardized questionnaire was sent to the physicians and medical institutions that reported the cases and controls selected by matching, and 90 cases and 371 controls were included in the main analysis.

**Table 2. Baseline characteristics of the cases and controls in the analysis dataset.**

| | Cases | Controls | |
| --- | --- | --- | --- |
| | No./total No. (%) | No./total No. (%) | P value |
| All | 90 (100) | 371 (100) | |
| Symptoms | | | |
| Vomiting | 50/87 (57.5) | 68/365 (18.6) | <0.001 |
| Diarrhea | 85/89 (95.5) | 355/370 (95.9) | 0.772 |
| Abdominal pain | 75/84 (89.3) | 288/347 (83.0) | 0.183 |
| Fever | 69/88 (78.4) | 140/365 (38.4) | <0.001 |
| Bloody stool | 74/89 (83.1) | 305/370 (82.4) | 1 |
| Mild | 12/61 (19.7) | 107/278 (38.5) | 0.005 |
| Moderate | 25/61 (41.0) | 134/278 (48.2) | 0.325 |
| Severe | 24/61 (39.3) | 37/278 (13.3) | <0.001 |
| STEC | | | |
| Positivity of stool culture | 54/90 (60.0) | 363/367 (98.9) | <0.001 |
| Shiga toxin | | | |
| Stx 1 | 23/62 (37.1) | 246/367 (67.0) | <0.001 |
| Stx 2 | 44/62 (71.0) | 210/367 (57.2) | 0.05 |
| Type unknown | 7/62 (11.3) | 36/367 (9.8) | 0.653 |
| Serotype | | | |
| O157 | 65/79 (82.3) | 208/365 (57.0) | <0.001 |
| O26 | 0/79 (0.0) | 89/365 (24.4) | <0.001 |
| O103 | 2/79 (2.5) | 13/365 (3.6) | 1 |
| O111 | 1/79 (1.3) | 15/365 (4.1) | 0.325 |
| Others | 11/79 (13.9) | 32/365 (8.8) | 0.205 |
| Anti-verotoxin antibody | 34/36 (94.4) | 23/30 (76.7) | 0.068 |
| Dialysis | 27/85 (31.8) | - | - |
| Clinical outcome | | | |
| Cured | 72/85 (84.7) | 354/357 (99.2) | <0.001 |
| Any complication | 10/85 (11.8) | 2/357 (0.6) | <0.001 |
| Encephalopathy | 13/89 (14.6) | - | - |
| Death | 3/85 (3.5) | 0/357 (0.0) | 0.007 |

STEC, Shiga toxin-producing *Escherichia coli*.

## The association between exposures and outcome

Fig 2 shows the association between treatment intervention (antibiotics, antidiarrheal agents, and probiotics) and primary outcome. In univariable analyses, any antibiotics use in children was significantly associated with a lower risk of HUS (mOR, 0.46 [95% CI 0.28–0.75]) although there was no significant association found in all ages and adults. This trend was similar but no significant association was found in multivariable analyses. Antidiarrheal agent use was significantly associated with a higher risk of HUS in all ages, and children both in univariable and multivariable analyses (mOR, 2.54 [1.37–4.72], 2.96 [1.43–6.12], mOR$_{adj}$ 2.07 [1.07–4.03], 2.65 [1.21–5.82], respectively). The use of probiotics was not associated with risk of HUS in any age group.

Fig 3 is the result of subgroup analyses showing the association between specific type of antibiotics and primary outcome. In univariable analyses, beta-lactam use was significantly associated with a higher risk of HUS in all age group (all ages mOR, 2.47 [95% CI 1.54–3.98], children 2.27 [1.29–4.02], adults 3.06 [1.26–7.46]). Fosfomycin was associated with a lower

**Table 3. Laboratory data of the cases and controls.**

| | Cases | | Controls | |
|---|---|---|---|---|
| | No. (%) | | No. (%) | |
| All | 90 (100) | | 371 (100) | |
| Blood test | 90 (100) | | 254 (68) | |
| | Mean (SD) | Missing data No. (%) | Mean (SD) | Missing data No. (%) |
| WBC, $10^3/\mu L$ | | | | |
| Initial | 14.5 (6.5) | 0 (0.0) | 10.35 (3.5) | 0 (0.0) |
| Worst | 20.87 (12.2) | 0 (0.0) | 12.06 (5.9) | 49 (19.3) |
| Hemoglobin, g/dL | | | | |
| Initial | 12.93 (2.6) | 0 (0.0) | 13.52 (1.4) | 2 (0.8) |
| Worst | 6.46 (1.4) | 0 (0.0) | 12.3 (1.9) | 50 (19.7) |
| Platelet, $\times 10^4/\mu L$ | | | | |
| Initial | 19.9 (13.3) | 0 (0.0) | 26.72 (8.5) | 3 (1.2) |
| Worst | 2.39 (2.1) | 0 (0.0) | 23.59 (8.8) | 51 (20.1) |
| CRP, mg/dL | | | | |
| Initial | 2.89 (3.7) | 0 (0.0) | 1.51 (2.7) | 6 (2.4) |
| Worst | 6.27 (6.8) | 0 (0.0) | 2.93 (5.4) | 54 (21.3) |
| BUN, mg/dL | | | | |
| Initial | 33.34 (40.1) | 1 (1.1) | 12.41 (7.8) | 30 (11.8) |
| Worst | 68.2 (39.6) | 0 (0.0) | 15.2 (14.2) | 64 (25.2) |
| Creatinine, mg/dL | | | | |
| Initial | 1.2 (2.1) | 0 (0.0) | 0.55 (0.7) | 30 (11.8) |
| Worst | 2.8 (2.6) | 0 (0.0) | 0.73 (1.4) | 63 (24.8) |
| Sodium, mEq/L | | | | |
| Initial | 135.2 (4.2) | 2 (2.2) | 138.9 (3.4) | 37 (14.6) |
| Worst | 132.2 (4.4) | 0 (0.0) | 137.3 (3.5) | 66 (26.0) |
| AST, IU/L | | | | |
| Initial | 47.1 (42.9) | 1 (1.1) | 26.52 (14.6) | 29 (11.4) |
| Worst | 123.7 (135.9) | 0 (0.0) | 34.36 (28.2) | 64 (25.2) |
| ALT, IU/L | | | | |
| Initial | 24.7 (27.4) | 1 (1.1) | 15.99 (9.3) | 29 (11.4) |
| Worst | 69.81 (63.8) | 0 (0.0) | 25.66 (31.1) | 64 (25.2) |

AST, aspartate aminotransferase; ALT, alanine aminotransferase; BUN, blood urea nitrogen, CRP, C-reactive protein; SD, standard deviation; WBC, white blood cell.

risk of HUS in all ages (mOR, 0.52 [0.33–0.81]) and in children (0.38 [0.23–0.62]) in univariable analyses although no significant association was found in multivariable analyses. Similar results were obtained in univariable and multivariable conditional logistic analyses among only patients detected with O157 (see S2 Table in S1 File).

In separate analyses of all ages, and in children, there was significant association between serotype O157 and the development of HUS by univariable conditional logistic regression analyses (see S3 Table in S1 File), although there was no significant association by multivariable conditional logistic regression analyses in each age groups.

Additionally, we also evaluated the time-related effect of fosfomycin administration on development of HUS within first five days of STEC infections. In the analysis of all ages, adults and children, there was no significant association between the timing of fosfomycin administration and development of HUS (see S4 Table in S1 File).

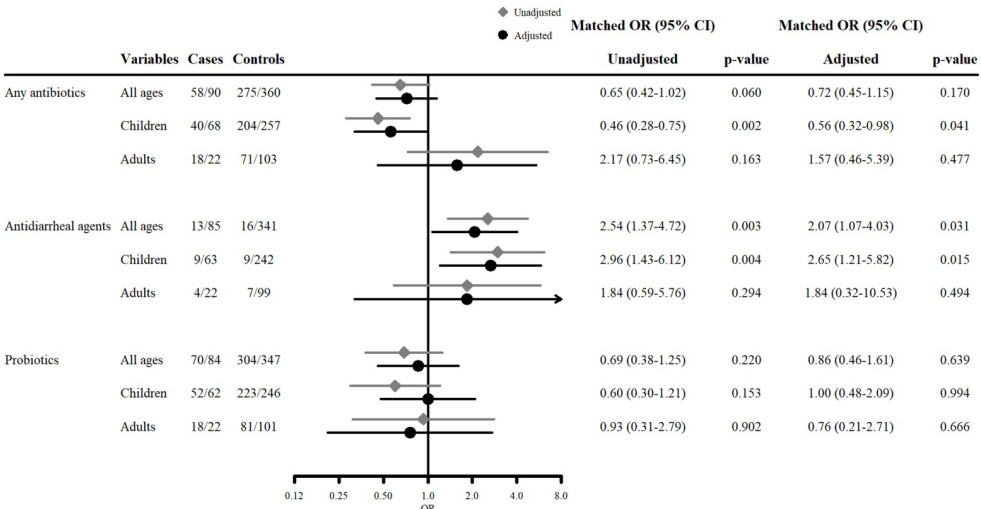

**Fig 2. The association between treatment and development of HUS.** Unadjusted matched odds ratios were calculated by univariable conditional logistic regression analysis. Adjusted matched odds ratios were calculated by multivariable conditional logistic regression analysis. The following covariates were used for the analyses of any antibiotics: age, sex, area, presence of bloody stool, initial white blood cell (WBC) count, initial CRP level, antidiarrheal agent use, and serotype O157. The following covariates were used for the analyses of antidiarrheal agents: age, sex, area, presence of bloody stool, initial WBC count, initial CRP level, serotype O157, and use of any antibiotics. The following covariates were used in the analysis of probiotics: age, sex, area, presence of bloody stool, initial WBC count, initial CRP level, antidiarrheal agent use, serotype O157, and any antibiotic use.

## Discussion

The current study results suggest that patients with STEC infection treated with antibiotics, particularly pediatric patients treated with fosfomycin, were at a lower risk of HUS. In

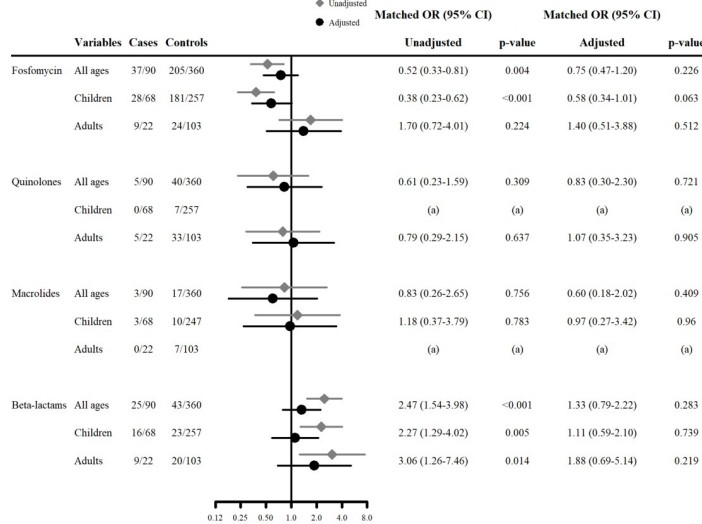

**Fig 3. The association between different type of antibiotics and development of HUS.** Unadjusted matched odds ratios were calculated by univariable conditional logistic regression analysis. Adjusted matched odds ratios were calculated by multivariable conditional logistic regression analysis. The following covariates were used for the analyses: age, sex, area, presence of bloody stool, initial white blood cell (WBC) count, initial CRP level, antidiarrheal agent use, and serotype O157. (a) In analyses for the association between quinolones and HUS in children and between macrolides and HUS in adults, the frequency of cases was zero and the odds ratio could not be properly estimated.

contrast, the use of beta-lactam antibiotics and antidiarrheal agents was significantly associated with a higher risk of HUS. These findings are in line with previous studies and might aid in understanding the discrepancy regarding the role of antibiotics for the treatment of STEC infections.

Although there has been much debate on the use of antibiotics for STEC infections, studies have repeatedly demonstrated the protective effect of fosfomycin against HUS mainly in children [19–22]. A recent review on the subject of antibiotic administration in patients with STEC infections included these Japanese studies and concluded that fosfomycin appears to be beneficial in these patients and may be able to avert HUS development, especially if administered early in the course of illness [28]. Although previous studies from Japan had important limitations such as selection bias, the estimates of the current study are more applicable to daily practice for several reasons. First, the present study results are based on the NESID, which encompasses all reported cases of STEC infection in Japan. Our dataset was representative of the NESID, indicating that the present study could adequately address the selection bias. Second, the presence of bloody stools to indicate severity was used as a matching factor and covariate in the statistical analyses. We assumed that clinicians were more likely to prescribe antibiotics for patients with severe disease, whereas most of the previous studies assessing the association of fosfomycin did not adequately address disease severity. Third, we used the internationally accepted diagnostic criteria for HUS [1]. A previous study suggested that the strength of association between antibiotics and development of HUS varied with case definition, which might even account for the discrepancy observed among the various studies [29]. In the present study, patients who did not strictly meet the diagnostic criteria for HUS but were clinically diagnosed with and treated for HUS were excluded from the analysis. The effect of fosfomycin was not clear in adults. The present study was underpowered to conduct a meaningful analysis in adults who are far less likely to develop HUS compared to children [2]. We were unable to demonstrate that early fosfomycin administration within five days of the onset of gastroenteritis symptoms reduced the risk of developing HUS (see S4 Table in S1 File). This might be due to the use of conditional logistic regression analysis, which resulted in a smaller number of samples belonging to the strata formed by matching factors. We adjusted by confounding factors such as age, gender, presence of bloody stool, initial WBC count and CRP because we considered physicians decide whether to administer antimicrobials in the early stages of disease only by these limited information. Therefore, it is not clear from our study whether fosfomycin should be administered in the early phase of STEC infection.

The present study confirmed that beta-lactams may have detrimental effects in patients with STEC infection, in agreement with previous studies [9, 17, 30, 31]. In a point of view of mechanism of action, both fosfomycin and beta-lactams are bactericidal, and act by inhibiting the synthesis of the peptidoglycan layer of bacterial cell walls. Several studies described the effect of antibiotic administration on toxin production in STEC infections. Class specific ability of certain antibiotics inducing phage replication and Shiga toxin release may explain conflicting results of the associations between different type of antibiotics and HUS development. Bacterial SOS response genes and Stx phage genes are known to be expressed together, and beta-lactams are associated with Stx2 expression in vitro as they are SOS-inducing antimicrobial agents, whereas fosfomycin are not [32]. Although these are plausible hypotheses, further investigation is needed to determine the mechanism by which each antimicrobial agent works in STEC infections. Although the current guidelines do not clearly distinguish risk according to the type of antibiotics, physicians should be aware that beta-lactams may be associated with the development of HUS.

Antidiarrheal agents were also associated with detrimental effects in patients with STEC infection, in agreement with previous studies [13, 14]. International and Japanese guidelines

for patients with infectious diarrhea clearly state that antidiarrheal drugs should not be used in patients with STEC infection because of the associated increase in the risk of HUS [33–36]. These known risk factors should be considered in daily practice. On the other hand, our result did not show significant association between probiotics and HUS development in any age group. As far as we know, there are no studies which analyze the association between probiotics for STEC patients and development of HUS. Recent study in children with diagnosis of acute intestinal infections, which showed no significant differences in prevention of moderate to severe diseases [37]. Studies demonstrating benefit are required to support current position papers [38] and expert opinions recommending use of early probiotics for STEC patients [15].

The present study has several limitations. First, our final analysis included less than half of the eligible cases and controls. However, the primary reason for exclusion was lack of response from each physician, which was expected due to the nature of paper-based questionnaires study. The number of cases enrolled was close to our a priori analysis that estimated sufficient power could be secured to detect OR with clinical significance in scenarios of 100 cases and 300 controls. Second, this was a retrospective observational study, and the possibility of additional confounding factors could not be ruled out. However, we did adjust for known risk factors of HUS such as the presence of bloody stool, which represents disease severity. Third, although the present study results might have high external validity for adaptation at least in Japan, differences in health care system or average day of presentation to health care services might confound and lead to different results in other countries. Of note, fosfomycin is used in a limited number of countries; it is though readily available in Europe for use in multi-drug resistant Gram-negative bacterial infections. Kakoullis *et al.* have argued that the beneficial effects of fosfomycin might represent a localized phenomenon because it is possible that the STEC strains endemic in Japan do not increase Shiga toxin release after fosfomycin exposure [28]. Fourth, case control studies are prone to recall bias in general. The data collection in our study was a retrospective chart review from participating physicians. They were asked to provide information on all antibiotics, antidiarrheal agents, and probiotics used during the course of the disease. Subsequently, only drugs used before the onset of HUS were counted as exposures in the cases. Therefore, recall bias by respondents was considered to be minimal.

In conclusion, although the present study did not show significant association between antibiotics administration and HUS development in the whole population, in the subgroup analysis, administration of fosfomycin for STEC infection in children younger than 15 years of age might be associated with a lower risk of HUS development. We also confirmed that beta-lactams and antimotility agents were associated with detrimental effects in patients with STEC infection. Future studies are warranted to establish tools for early diagnosis of STEC gastroenteritis to initiate optimal treatment and to prospectively monitor for the development of HUS.

## Supporting information

**S1 File.**
(DOCX)

**S1 Dataset.**
(XLSX)

## Acknowledgments

We thank the individuals and institutions that contributed data to the study.

## Author Contributions

**Conceptualization:** Shota Myojin, Mayumi Sako, Tohru Kobayashi, Isao Miyairi.

**Data curation:** Shota Myojin, Mayumi Sako, Tohru Kobayashi, Isao Miyairi.

**Formal analysis:** Shota Myojin.

**Funding acquisition:** Takashi Igarashi.

**Investigation:** Shota Myojin, Isao Miyairi.

**Methodology:** Shota Myojin, Kyongsun Pak, Mayumi Sako, Tohru Kobayashi, Takuri Takahashi, Tomimasa Sunagawa, Norihiko Tsuboi, Kenji Ishikura, Masaya Kubota, Mitsuru Kubota, Ichiro Morioka, Isao Miyairi.

**Project administration:** Shota Myojin, Mayumi Sako, Tohru Kobayashi, Isao Miyairi.

**Resources:** Takuri Takahashi, Tomimasa Sunagawa.

**Supervision:** Mayumi Sako, Tohru Kobayashi, Ichiro Morioka, Isao Miyairi.

**Validation:** Isao Miyairi.

**Visualization:** Shota Myojin, Kyongsun Pak.

**Writing – original draft:** Shota Myojin.

**Writing – review & editing:** Mayumi Sako, Tohru Kobayashi, Isao Miyairi.

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
