## [Decision Letter · Decision Letter 0]

3 Nov 2021

PONE-D-21-32011

Interventions for Shiga toxin-producing Escherichia coli gastroenteritis and risk of hemolytic uremic syndrome: a population-based matched case control study

PLOS ONE

Dear Dr. Miyairi,

Thank you for submitting your manuscript to PLOS ONE. After careful consideration, we feel that it has merit but does not fully meet PLOS ONE’s publication criteria as it currently stands. Therefore, we invite you to submit a revised version of the manuscript that addresses the points raised during the review process.

The manuscript focuses on a topic of potential interest. However, the manuscript has several shortcomings that preclude sound conclusions. To mention some of them, i) concern about the novelty of the study; ii) need to address in the discussion the nature of bactericidal versus bacteriostatic antibiotics; iii) need to provide in the tables the use of anti-diarrheal agents and antibiotics within distinct groups of patients; iv) unclear whether in cases treatments were counted only when they were administered before the development of HUS and whether this was comparable with the control group; v) need to elaborate on the time window of fosfomycin treatment  in children with STEC-HUS, and to clarify which  is the effect of fosfomycin at later time points of treatment; vi) unclear in figure 1 what is the reason for exclusion of 6668 not matched controls; vii) concern about the fact that in Table 2 cases and control cohorts are not always comparable; viii) unclear whether the observed associations are also present when comparing Stx2 infected cohorts or LPS O157 positive cohorts; ix) concern about the fact that authors’ data do not support the statement reported in lines 328-329; x) need to elaborate on the molecular mechanism of the beneficial effect of fosfomycin in STEC-HUS patients; xi) unclear whether there is any information available about the excluded patients to know if they are systematically different from those patients who were included in the analysis; xii) unclear whether the use of antibiotics look bad because they are given to sicker people, or they are bad even in equivalently sick people; xiii) need to clarify their interpretation of the difference between the unadjusted and adjusted ORs; xiv) concern about the fact that beta lactams are actually worse in adults.

We look forward to receiving your revised manuscript.

Kind regards,

Giuseppe Remuzzi

Academic Editor

PLOS ONE

Journal Requirements:

Reviewers' comments:

Reviewer's Responses to Questions

Comments to the Author

1. Is the manuscript technically sound, and do the data support the conclusions?

Reviewer #1: Yes

Reviewer #2: Yes

Reviewer #3: Yes

2. Has the statistical analysis been performed appropriately and rigorously?

Reviewer #1: Yes

Reviewer #2: I Don't Know

Reviewer #3: Yes

3. Have the authors made all data underlying the findings in their manuscript fully available?

Reviewer #1: Yes

Reviewer #2: Yes

Reviewer #3: Yes

4. Is the manuscript presented in an intelligible fashion and written in standard English?

Reviewer #1: Yes

Reviewer #2: Yes

Reviewer #3: Yes

5. Review Comments to the Author

Reviewer #1: PLOS One manuscript STEC-HUS and fosfomycin:

The manuscript by Myojin et al. entitled “Intervention for Shiga toxin producing Escherichia coli gastroenteritis and risk of haemolytic uremic syndrome: a population-based matched case control study (reference PONE-D-21-32011)” reports on the role of antibiotics in the treatment of Shiga toxin-producing E. Coli (STEC) infections. Authors evaluate the association between treatment (antibiotics, antidiarrheal agents and probiotics) for STEC infection and HUS development. A population based matched case-control study was performed with the data from NESID from 2017 and 2018. In this period 7760 patients with STEC infection were registered. The main analysis was executed with 90 patients with HUS and 371 patients without HUS. The patient cohort is impressive and the study has been performed adequately. Important information for the clinical care of STEC HUS patients is reported. Antibiotics (especially fosfomycin) may prevent the development of HUS in children and the use of antidiarrheal agents should be avoided. Next issues need to be addressed:

1.)Line 179: In cases treatments were counted only when they were administered before the development of HUS? Was this comparable with the control group? This may include bias in the data set?

2.)Authors should elaborate on the time window of fosfomycin treatment in children with STEC-HUS? What is the effect of fosfomycin at later time points of treatment?

3.)Figure 1: What is the reason for exclusion of “6668 not matched controls”. Please specify.

4.)Table 1: You describe 182 HUS patients. In the age-distribution part only 180 patients are reported for all ages? Two patients are lacking?

5.)Table 2: Cases and control cohorts are not always comparable (for example for Stx1/Stx2 in stool, serotypes 0157 and so on). Are observed associations also present when comparing Stx2 infected cohorts or LPS 0157 positive cohorts?

6.)Line 328-329: “fosfomycin appears to be beneficial in STEC HUS patients and may avert HUS development, especially if administered early in course of illness”. Do your data also support this statement? Can you improve your illustration to support this statement?

7.)The authors should elaborate on the molecular mechanism of the beneficial effect of fosfomycin in STEC-HUS patients (see also reference 26).

Advise: Moderate revision.

Reviewer #2: The authors studied the association of several factors with the occurrence of HUS in the setting of STEC infection. Their analysis focused on the use of antidiarrheal drugs and antibiotics. They show that the use of antidiarrheal and the b-lactams increase the risk of HUS in this setting whereas fosfomycin decreases this risk. The main issue I have with this study is its rather limited novelty, as several previous studies had assessed this question, as stated by the authors.

a) The nature of bactericidal versus bacteriostatic antibiotics should be addressed in the discussion, as the lysis of bacteria (in contrast to only inhibition of bacterial growth) may contribute to the spread of verotoxin and hence to the onset of HUS.

b) The use of antidiarrheal agents and antibiotics within distinct groups of patients is not shown in the tables.

Reviewer #3: A very interesting and well written paper. The early treatment of HUS is a clinically important topic.

As this is a reportable condition in Japan, the likelihood of there being a large number of unreported cases that would be missing from this type of analysis is low. The conclusions are supported by the data, but are hampered by the small sample size. In particular, as the authors point out, about half of the HUS cases were not included in the study, mostly due to lack of response from the physician. My question is whether there is any information available about these excluded patients to know if they are systematically different from those patients who were included in the analysis?

It is interesting that the OR for the use of the individual classes of antibiotics tend to lose their significance in the adjusted analyses. My understanding of why the authors did these analyses was to try to blunt the effect of the presenting characteristics that influence the physician decision to give antibiotics (in other words, does the use of antibiotics look bad because they are given to sicker people, or are they bad even in equivalently sick people). The authors need to be clearer about their interpretation of the difference between the unadjusted and adjusted ORs. For example, the difference in the adjusted and unadjusted ORs for beta lactam use in adults looks important to me. I’m not convinced that they are actually worse in adults. The authors should be clear about which analysis they put the most weight upon. I don’t see that same difference in adjusted and non-adjusted ORs for antidiarrheal agents, which generally look bad by either analytic technique.

6. PLOS authors have the option to publish the peer review history of their article (what does this mean?). If published, this will include your full peer review and any attached files.

Do you want your identity to be public for this peer review? For information about this choice, including consent withdrawal, please see our Privacy Policy.

Reviewer #1: No

Reviewer #2: No

Reviewer #3: No

---

## [Author Response · Author response to Decision Letter 0]

3 Dec 2021

Response to reviewers and editors

We wish to express our appreciation to the reviewers and editors for their insightful comments, which have helped us significantly improve the paper. We have addressed your comments with point-by-point responses, and revised the manuscript accordingly.

Reviewer #1

Comment 1: Line 179: In cases treatments were counted only when they were administered before the development of HUS? Was this comparable with the control group? This may include bias in the data set?

Response 1: We thank the Reviewer for this pertinent comment. As we described in “Outcome, exposures, and potential confounders” section (Lines 179-180), treatments were counted only when they were administered before development of HUS in cases. Due to the nature of the case control study, controls did not receive any exposure of interest. In general, case control studies are prone to recall bias. The data collection in our study was a retrospective chart review from participating physicians. They were asked to provide information on all antibiotics, antidiarrheals, and probiotics used during the course of the disease. Subsequently, only drugs used before the onset of HUS were counted as exposures in the cases. Therefore, recall bias by respondents was considered to be minimal.

We have added following sentences as one of the limitations (Lines 409-414).

“Fourth, case control studies are prone to recall bias in general. The data collection in our study was a retrospective chart review from participating physicians. They were asked to provide information on all antibiotics, antidiarrheal agents, and probiotics used during the course of the disease. Subsequently, only drugs used before the onset of HUS were counted as exposures in the cases. Therefore, recall bias by respondents was considered to be minimal.”

Comment 2: Authors should elaborate on the time window of fosfomycin treatment in children with STEC-HUS? What is the effect of fosfomycin at later time points of treatment?

Response 2: A previous study that examined the causal effects of fosfomycin on the development of HUS concluded that the risk was reduced when fosfomycin was administered at an earlier phase of illness (as we explained in Lines 333-336). For this reason, we also examined the association between the administration of fosfomycin within five days of onset and the development of HUS, but we did not include the results in our first submission. This was because we did not calculate the sample size beforehand for this subgroup analysis, so we considered it difficult to interpret the results. Based on the reviewer's comments, we have added the results of the analysis to Table S4 (shown below). In order to avoid deriving arbitrary results, the covariates used in the multivariate analysis are the same as those used in the main analysis. In each age group, there was no statistically significant result in either univariate or multivariate analysis. It is difficult to discuss the effectiveness of early administration of fosfomycin in this study, and we consider that further research is necessary.

“Table S4. Univariable and multivariable conditional logistic regression analysis to assess the effect of the timing of fosfomycin use within five days of illness on HUS development.

 Matched OR (95% CI) Matched OR (95% CI)

 Cases Controls Unadjusted P value Adjusted P value

All ages 32/90 147/371 0.76 (0.48-1.19) 0.227 0.85 (0.53-1.36) 0.499

Children 24/68 125/266 0.67 (0.41-1.11) 0.124 0.77 (0.45-1.29) 0.315

Adults 6/22 20/105 1.23 (0.47-3.20) 0.673 1.19 (0.38-3.76) 0.764

Unadjusted matched odds ratios were calculated by univariable conditional logistic regression analysis. Adjusted matched odds ratios were calculated by multivariable conditional logistic regression analysis using the following covariates: age, sex, area, presence of bloody stool, initial white blood cell count, initial C-reactive protein level.

CI, confidence interval; HUS, hemolytic uremic syndrome; OR, odds ratio”

We have added a reference to Table S4 in “Results” section as follows (Lines 318-321).

“Additionally, we also evaluated the time-related effect of fosfomycin administration on development of HUS within first five days of STEC infections. In the analysis of all ages, adults and children, there was no significant association between the timing of fosfomycin administration and development of HUS (see Table S4).”

Comment 3: Figure 1: What is the reason for exclusion of “6668 not matched controls”. Please specify.

Response 3: This study is a matched case control study. As we described in the section of “Data collection”, cases were selected from patients with a record of HUS diagnosis in the NESID, and controls were selected from patients without a record of HUS who were matched to cases with a case-control ratio of 1:5. Therefore, 6668 were not matched as controls. We have modified Figure 1 to make this flow easier to understand from:

to

Comment 4: Table 1: You describe 182 HUS patients. In the age-distribution part only 180 patients are reported for all ages? Two patients are lacking?

Response 4: We have checked the original data and found that it was a transcription error for the age group 0-6, so we have corrected it. The following table underlines the part that has been changed. We appreciate the reviewer pointing this out.

Table 1 (modified). 

 Group, no. (%)

 NESID, n (%) Matched controls, n (%) Analysis dataset, n (%)

 HUS Non-HUS Cases Controls 

All 182 (100) 7578 (100) 910 (100) 90 (100) 371 (100)

Sex 

 Female 114 (62.6) 4231 (55.8) 570 (62.6) 57 (63.3) 213 (57.4)

Age, years 

 0–6 92 (50.5) 1540 (20.3) 460 (50.5) 50 (55.6) 196 (52.8)

 7–15 28 (15.4) 871 (11.5) 140 (15.4) 18 (20.0) 70 (18.9)

 16–64 41 (22.5) 4173 (55.1) 205 (22.5) 18 (20.0) 72 (19.4)

 ≥65 21 (11.5) 994 (13.1) 105 (11.5) 4 (4.4) 33 (8.9)

Area 

 Hokkaido/Tohoku 18 (9.9) 1179 (15.6) 35 (3.8) 8 (8.9) 10 (2.7)

 Kanto 86 (47.3) 2807 (37.0) 98 (10.8) 41 (45.6) 43 (11.6)

 Chubu 31 (17.0) 1340 (17.7) 92 (10.1) 19 (21.1) 35 (9.4)

 Kinki 27 (14.8) 893 (11.8) 217 (23.8) 11 (12.2) 96 (25.9)

 Chugoku/Shikoku 4 (2.2) 501 (6.6) 113 (12.4) 3 (3.3) 64 (17.3)

 Kyushu/Okinawa 1 (8.8) 858 (11.3) 355 (39.0) 8 (8.9) 123 (33.2)

Comment 5: Table 2: Cases and control cohorts are not always comparable (for example for Stx1/Stx2 in stool, serotypes 0157 and so on). Are observed associations also present when comparing Stx2 infected cohorts or LPS 0157 positive cohorts?

Response 5: The subgroup analysis of only O157 positive cohorts is described in Table S3 in the original manuscript. Analysis of the Stx2 positive cohort was performed, however, we have not shown the results because we judged it to be difficult to interpret the result for the following reasons. First of all, we were not able to create a subgroup of Stx2 positive patients because the notification based on the Infectious Diseases Control Law in Japan includes the following options: Stx1, Stx2, Stx1 plus 2, and serotype unknown. Next, there were many missing data for Stx. Finally, it was difficult to interpret the results due to the small sample size of the subgroup analysis.

Comment 6: Line 328-329: “fosfomycin appears to be beneficial in STEC HUS patients and may avert HUS development, especially if administered early in course of illness”. Do your data also support this statement? Can you improve your illustration to support this statement?

Response 6:

As we described in Response 2, we analyzed the association between early fosfomycin administration and HUS development, however there were no statistically significant results. We have therefore added the following text in “Discussion” (Line 353-362):

“We were unable to demonstrate that early fosfomycin administration within two or five days of the onset of gastroenteritis symptoms reduced the risk of developing HUS (see Table S4). This might be due to the use of conditional logistic regression analysis, which resulted in a smaller number of samples belonging to the strata formed by matching factors. We adjusted by confounding factors such as age, gender, presence of bloody stool, initial WBC count and CRP because we considered physicians decide whether to administer antimicrobials in the early stages of disease only by these limited information. Therefore, it is not clear from our study whether fosfomycin should be administered in the early phase of STEC infection.”

Comment 7: The authors should elaborate on the molecular mechanism of the beneficial effect of fosfomycin in STEC-HUS patients (see also reference 26).

Response 7:

The reasons for the different results for the different types of antimicrobial agents may be due to the influence of differences in the action of antimicrobial agents in the human body. As pointed out by the reviewer, it is important to consider how the molecular mechanisms of different antibiotics such as beta-lactams and fosfomycin differ. For this reason, we have added the following text to “Discussion“ (Line 366-376).

“In a point of view of mechanism of action, both fosfomycin and beta-lactams are bactericidal, and act by inhibiting the synthesis of the peptidoglycan layer of bacterial cell walls. Several studies described the effect of antibiotic administration on toxin production in STEC infections. Class specific ability of certain antibiotics inducing phage replication and Shiga toxin release may explain conflicting results of the associations between different type of antibiotics and HUS development. Bacterial SOS response genes and Stx phage genes are known to be expressed together, and beta-lactams are associated with Stx2 expression in vitro as they are SOS-inducing antimicrobial agents, whereas fosfomycin are not. Although these are plausible hypotheses, further investigation is needed to determine the mechanism by which each antimicrobial agent works in STEC infections.”

We have also the following reference.

“32. Joseph A, Cointe A, Mariani KP, et al. Shiga Toxin-Associated Hemolytic Uremic Syndrome: A Narrative Review. Toxins (Basel). 2020;12(2):67. doi: 10.3390/toxins12020067. “

Reviewer #2

Comment 8: The nature of bactericidal versus bacteriostatic antibiotics should be addressed in the discussion, as the lysis of bacteria (in contrast to only inhibition of bacterial growth) may contribute to the spread of verotoxin and hence to the onset of HUS.

Response 8: We appreciate the Reviewer’s comment on this point. The reasons for the different results for the different types of antimicrobial agents may be due to the influence of differences in the action of antimicrobial agents in the human body. As pointed out by the reviewer, it is important to consider how the molecular mechanisms of different antibiotics such as beta-lactams and fosfomycin differ. For this reason, we have added the following text to the Discussion (Line 366-376).

“In a point of view of mechanism of action, both fosfomycin and beta-lactams are bactericidal, and act by inhibiting the synthesis of the peptidoglycan layer of bacterial cell walls. Several studies described the effect of antibiotic administration on toxin production in STEC infections. Class specific ability of certain antibiotics inducing phage replication and Shiga toxin release may explain conflicting results of the associations between different type of antibiotics and HUS development. Bacterial SOS response genes and Stx phage genes are known to be expressed together, and beta-lactams are associated with Stx2 expression in vitro as they are SOS-inducing antimicrobial agents, whereas fosfomycin are not. Although these are plausible hypotheses, further investigation is needed to determine the mechanism by which each antimicrobial agent works in STEC infections.”

We have also added the following reference.

“32. Joseph A, Cointe A, Mariani KP, et al. Shiga Toxin-Associated Hemolytic Uremic Syndrome: A Narrative Review. Toxins (Basel). 2020;12(2):67. doi: 10.3390/toxins12020067. “

Comment 9: The use of antidiarrheal agents and antibiotics within distinct groups of patients is not shown in the tables.

Response 9: We are appreciated to the reviewer’s suggestion. In the original manuscript, we have showed the distribution of the exposures of interest (any antibiotics, antidiarrheal agents, and probiotics) for all age groups, children, and adults in Figure 2. Similarly, use by antimicrobial class is summarized in Figure 3.

Reviewer #3

Comment 10: As this is a reportable condition in Japan, the likelihood of there being a large number of unreported cases that would be missing from this type of analysis is low. The conclusions are supported by the data, but are hampered by the small sample size. In particular, as the authors point out, about half of the HUS cases were not included in the study, mostly due to lack of response from the physician. My question is whether there is any information available about these excluded patients to know if they are systematically different from those patients who were included in the analysis?

Response 10: We thank the Reviewer for this insightful comment. Since all STEC infections are notifiable in Japan, we agree with the Reviewer's comment that there are probably not many cases that are not reported. It is important for the interpretation of the analysis results that the demographic data of the patients excluded for reasons such as not returning the questionnaires are not significantly different from the demographic data of the patients included in the analysis. The proportions of sex, age, and region of patients with HUS in the analyzed dataset were similar to those of patients with HUS registered in the NESID. This means the baseline characteristics of analyzed dataset is similar to the original data of NESID (as shown in Table 1). We have added following descriptions in the manuscript (Lines 256-257).

“Although about half of the cases were not included mostly due to lack of responses, the proportions of sex, age, and region of patients with HUS in the analyzed dataset were similar to those of patients with HUS registered in the NESID.”

Comment 11: It is interesting that the OR for the use of the individual classes of antibiotics tend to lose their significance in the adjusted analyses. My understanding of why the authors did these analyses was to try to blunt the effect of the presenting characteristics that influence the physician decision to give antibiotics (in other words, does the use of antibiotics look bad because they are given to sicker people, or are they bad even in equivalently sick people). The authors need to be clearer about their interpretation of the difference between the unadjusted and adjusted ORs. For example, the difference in the adjusted and unadjusted ORs for beta lactam use in adults looks important to me. I’m not convinced that they are actually worse in adults. The authors should be clear about which analysis they put the most weight upon. I don’t see that same difference in adjusted and non-adjusted ORs for antidiarrheal agents, which generally look bad by either analytic technique.

Response 11: We believe that previous studies examining the association between antimicrobial agents and HUS have insufficiently considered the confounding factor of patient severity. In other words, it is important to adjust for the severity of disease, because it is possible that patients with more severe disease initially received antimicrobial agents more frequently, and it is also possible that patients with more severe disease were more likely to develop HUS. Therefore, we used the presence of bloody stool as one of the covariates in the multivariable conditional logistic regression analysis to reflect the severity of disease. As the reviewer commented, there is difference between unadjusted ORs and adjusted ORs, and this was more noticeable in beta-lactams. The following tables show the distribution of covariates used in the multivariate analysis for all ages, children, and adults with and without beta-lactams (Rev-Table 1-3). In all age groups, the beta-lactam group had higher rates of three known risk factors for HUS: white blood cells, CRP, and O-157. This suggests that the ORs that were significant in the univariate analysis became insignificant after adjustment for confounding factors including these factors.

Rev-Table 1. The distribution of covariates used in the multivariate analysis for all ages with and without beta-lactams.

 All Ages

 with beta-lactams

(n=68) without beta-lactams

 (n=382)

Age 0-6 27/68 39.7% 212/382 55.5%

 7-15 12/68 17.6% 74/382 19.4%

 16-64 16/68 23.5% 72/382 18.8%

 ≥65 13/68 19.1% 24/382 6.3%

Sex Female 46/68 67.6% 217/382 56.8%

Area Hokkaido/Tohoku 5/68 7.4% 12/382 3.1%

 Kanto 13/68 19.1% 69/382 18.1%

 Chubu 8/68 11.8% 45/382 11.8%

 Kinki 20/68 29.4% 85/382 22.3%

 Chugoku/Shikoku 8/68 11.8% 57/382 14.9%

 Kyushu/Okinawa 14/68 20.6% 114/382 29.8%

Bloody Stool 60/68 88.2% 310/382 81.2%

WBC>10000/μL 41/64 64.1% 142/276 51.4%

CRP>1.2mg/dL 38/63 60.3% 97/271 35.8%

O157 52/68 76.5% 221/382 57.9%

Antidiarrheal agents 5/65 7.7% 24/361 6.6%

Rev-Table 2. The distribution of covariates used in the multivariate analysis for children with and without beta-lactams.

 Children

 with beta-lactams

(n=68) without beta-lactams 

(n=382)

Age 0-6 27/39 69.2% 212/286 74.1%

 7-15 12/39 30.8% 74/286 25.9%

 16-64 0/39 0.0% 0/286 0.0%

 ≥65 0/39 0.0% 0/286 0.0%

Sex Female 19/39 48.7% 131/286 45.8%

Area Hokkaido/Tohoku 6/39 15.4% 9/286 3.1%

 Kanto 10/39 25.6% 62/286 21.7%

 Chubu 5/39 12.8% 34/286 11.9%

 Kinki 9/39 23.1% 54/286 18.9%

 Chugoku/Shikoku 5/39 12.8% 38/286 13.3%

 Kyushu/Okinawa 8/39 20.5% 89/286 31.1%

Bloody Stool 35/39 89.7% 224/286 78.3%

WBC>10000/μL 24/36 66.7% 99/187 52.9%

CRP>1.2mg/dL 17/36 47.2% 54/185 29.2%

O157 27/39 69.2% 153/286 53.5%

Antidiarrheal agents 3/37 8.1% 15/267 5.6%

Rev-Table 3. The distribution of covariates used in the multivariate analysis for adults with and without beta-lactams.

 Adults

 with beta-lactams

 (n=68) without beta-lactams

(n=382)

Age 0-6 0/29 0.0% 0/96 0.0%

 7-15 0/29 0.0% 0/96 0.0%

 16-64 16/29 55.2% 72/96 75.0%

 ≥65 13/29 44.8% 24/96 25.0%

Sex Female 27/29 93.1% 86/96 89.6%

Area Hokkaido/Tohoku 3/29 10.3% 3/96 3.1%

 Kanto 3/29 10.3% 7/96 7.3%

 Chubu 3/29 10.3% 11/96 11.5%

 Kinki 11/29 37.9% 31/96 32.3%

 Chugoku/Shikoku 3/29 10.3% 19/96 19.8%

 Kyushu/Okinawa 6/29 20.7% 25/96 26.0%

Bloody Stool 25/29 86.2% 86/96 89.6%

WBC>10000/μL 17/29 60.7% 43/89 48.3%

CRP>1.2mg/dL 21/27 77.8% 43/86 50.0%

O157 25/29 86.2% 68/96 70.8%

Antidiarrheal agents 2/27 7.4% 9/94 9.6%

From these additional considerations, we have modified the following texts regarding beta-lactams in “Discussion” section from:

“The present study confirmed that beta-lactams were associated with detrimental effects in patients with STEC infection, in agreement with previous studies.”

to

“The present study confirmed that beta-lactams may have detrimental effects in patients with STEC infection, in agreement with previous studies.” (Lines 364-365)

and from:

“Although the current guidelines do not clearly distinguish risk according to the type of antibiotics, physicians should be aware that beta-lactams are associated with the development of HUS.”

to

“Although the current guidelines do not clearly distinguish risk according to the type of antibiotics, physicians should be aware that beta-lactams may be associated with the development of HUS.” (Line 376-378)

We wish to thank the Reviewer again for his or her valuable comments.

Isao Miyairi, MD, PhD

Department of Pediatrics, Hamamatsu University School of Medicine

1-20-1, Handayama, Higashiku, Hamamatsu, Shizuoka 431-3192, Japan

+81-53-435-2312 (telephone)

+81-53-431-2311 (fax)

miyairi@hama-med.ac.jp

---

## [Decision Letter · Decision Letter 1]

18 Jan 2022

Interventions for Shiga toxin-producing Escherichia coli gastroenteritis and risk of hemolytic uremic syndrome: a population-based matched case control study

PONE-D-21-32011R1

Dear Dr. Miyairi,

We’re pleased to inform you that your manuscript has been judged scientifically suitable for publication and will be formally accepted for publication once it meets all outstanding technical requirements.

**The revised version of the manuscript is definitely improved. The authors have properly addressed the reviewers’ comments.**

Kind regards,

Giuseppe Remuzzi

Academic Editor

PLOS ONE

Additional Editor Comments (optional):

Reviewers' comments:

Reviewer's Responses to Questions

**Comments to the Author**

1. If the authors have adequately addressed your comments raised in a previous round of review and you feel that this manuscript is now acceptable for publication, you may indicate that here to bypass the “Comments to the Author” section, enter your conflict of interest statement in the “Confidential to Editor” section, and submit your "Accept" recommendation.

Reviewer #1: (No Response)

Reviewer #2: All comments have been addressed

2. Is the manuscript technically sound, and do the data support the conclusions?

Reviewer #1: Yes

Reviewer #2: (No Response)

3. Has the statistical analysis been performed appropriately and rigorously? 

Reviewer #1: Yes

Reviewer #2: I Don't Know

4. Have the authors made all data underlying the findings in their manuscript fully available?

Reviewer #1: Yes

Reviewer #2: Yes

5. Is the manuscript presented in an intelligible fashion and written in standard English?

Reviewer #1: Yes

Reviewer #2: Yes

6. Review Comments to the Author

Reviewer #1: In the revised manuscript by Myojin et al. entitled “Intervations for Shiga toxin-producing Escherichia coli gastroenteritis and risk of haemolytic uremic syndrome: a population-based matched case control study (reference PONE-D-21-3201R1)”

The authors responded well to the raised issues for the first manuscript. They performed substantial adaptations in the result plus discussion section of the revised manuscript. I think the revised version improved substantially. For me in the current form this manuscript is acceptable for publication in a future issue of PLOS ONE.

Reviewer #2: The authors have adressed most of the comments and the manuscript has been improved. I have no further comments.

7. PLOS authors have the option to publish the peer review history of their article (what does this mean?). If published, this will include your full peer review and any attached files.

Reviewer #1: No

Reviewer #2: No

---

## [Editor Report · Acceptance letter]

28 Jan 2022

PONE-D-21-32011R1 

Interventions for Shiga toxin-producing *Escherichia coli* gastroenteritis and risk of hemolytic uremic syndrome: a population-based matched case control study 

Dear Dr. Miyairi:

I'm pleased to inform you that your manuscript has been deemed suitable for publication in PLOS ONE. Congratulations! Your manuscript is now with our production department. 

Kind regards, 

on behalf of

Prof. Giuseppe Remuzzi 

Academic Editor

PLOS ONE